# DNA methylation in *Arabidopsis* has a genetic basis and shows evidence of local adaptation

Manu J Dubin[1]*[†], Pei Zhang[1,2][†], Dazhe Meng[1,2][†], Marie-Stanislas Remigereau[2][†], Edward J Osborne[3], Francesco Paolo Casale[4], Philipp Drewe[5,6], André Kahles[5,6], Geraldine Jean[5,6], Bjarni Vilhjálmsson[1], Joanna Jagoda[1], Selen Irez[1], Viktor Voronin[1], Qiang Song[2], Quan Long[1], Gunnar Rätsch[5,6], Oliver Stegle[4], Richard M Clark[3,7], Magnus Nordborg[1,2]*

[1]Gregor Mendel Institute, Austrian Academy of Sciences, Vienna Biocenter, Vienna, Austria; [2]Molecular and Computational Biology, University of Southern California, Los Angeles, United States; [3]Department of Biology, University of Utah, Salt Lake City, United States; [4]European Molecular Biology Laboratory, European Bioinformatics Institute, Wellcome Trust Genome Campus, Cambridge, United Kingdom; [5]Friedrich Miescher Laboratory, Max Planck Society, Tübingen, Germany; [6]Memorial Sloan-Kettering Cancer Center, New York, United States; [7]Center for Cell and Genome Science, University of Utah, Salt Lake City, United States

*For correspondence: manu.dubin@gmi.oeaw.ac.at (MJD); magnus.nordborg@gmi.oeaw.ac.at (MN)

[†]These authors contributed equally to this work

Competing interests: The authors declare that no competing interests exist.

**Abstract** Epigenome modulation potentially provides a mechanism for organisms to adapt, within and between generations. However, neither the extent to which this occurs, nor the mechanisms involved are known. Here we investigate DNA methylation variation in Swedish *Arabidopsis thaliana* accessions grown at two different temperatures. Environmental effects were limited to transposons, where CHH methylation was found to increase with temperature. Genome-wide association studies (GWAS) revealed that the extensive CHH methylation variation was strongly associated with genetic variants in both *cis* and *trans*, including a major *trans*-association close to the DNA methyltransferase CMT2. Unlike CHH methylation, CpG gene body methylation (GBM) was not affected by growth temperature, but was instead correlated with the latitude of origin. Accessions from colder regions had higher levels of GBM for a significant fraction of the genome, and this was associated with increased transcription for the genes affected. GWAS revealed that this effect was largely due to *trans*-acting loci, many of which showed evidence of local adaptation.

## Main

To better understand how genotype and environment interact to affect DNA methylation and transcription, we grew 150 *Arabidopsis thaliana* accessions from Sweden (*Long et al., 2013*) in two different environments, 10°C and 16°C, chosen because they lead to very different flowering behavior (*Atwell et al., 2010*). Relying on existing genome sequence information (*Long et al., 2013*), methylome- and transcriptome-sequencing data were generated (see 'Materials and methods').

In plants, DNA methylation occurs on cytosines in the CG, CHG, and CHH contexts (where H is any nucleotide except for C), each of which is catalyzed by independent pathways (*Finnegan et al., 1998*; *Stroud et al., 2014*). Consistent with previous results (*Vaughn et al., 2007*; *Eichten et al., 2013*; *Schmitz et al., 2013*; *Li et al., 2014*; *Seymour et al., 2014*; *Hagmann et al., 2015*) we found considerable variation between accessions regardless of context, even at the level of genome-wide averages (*Figure 1A*). Temperature, on the other hand, did not appear to affect genome-wide CG or

**eLife digest** Organisms need to adapt quickly to changes in their environment. Mutations in the DNA sequence of genes can lead to new adaptations, but this can take many generations. Instead, altering how genes are switched on by changing how the DNA is packaged in cells can allow organisms to adapt within and between generations. One way that genes are controlled in organisms is by a process known as DNA methylation, where 'methyl' tags are added to DNA and act as markers for other proteins involved in activating genes.

DNA is made of four different molecules called 'nucleotides' that are arranged in different orders to produce a vast variety of DNA sequences. One type of DNA methylation can happen at sites where a nucleotide called cytosine is followed by two other non-cytosine nucleotides. Another type of methylation can take place at sites where a cytosine is followed by a guanine nucleotide. However, it is not clear how big a role DNA methylation plays in allowing organisms to adapt to their changing environment.

Here, Dubin, Zhang, Meng, Remigereau et al. studied DNA methylation in a plant called *Arabidopsis thaliana*. Several different varieties of *A. thaliana* plants from Sweden were grown at two different temperatures. The experiments showed that the *A. thaliana* plants grown at higher temperatures were more likely to have methyl tags attached to sections of DNA called transposons, which are able to move around the genome. There was a lot of variety in the levels of this DNA methylation in the different plants, and some of it was shown to be associated with variation in a gene that is involved in DNA methylation.

However, not all of the DNA methylation in these plants was sensitive to the temperature the plants were grown in. Dubin, Zhang, Meng, Remigereau et al. show that the pattern of a type of DNA methylation that is found within genes depends on how far north in Sweden the plants' ancestors came from rather than the temperature the plants were grown in. Plants that originated from colder regions, farther north, had more DNA methylation within many genes and these genes were more active.

These findings suggest that genetic differences in these plants strongly influence the levels of DNA methylation, and they provide the first direct link between DNA methylation and adaption to the environment. Future studies should reveal how DNA methylation is regulated in these plants, and whether it plays a key role in adaptation, or merely reflects other changes in the genome.

CHG methylation, but had a significant effect on CHH methylation, levels of which were 14% higher at 16°C than at 10°C, on average (*Figure 1A*). To investigate the genetic basis of DNA methylation, we performed genome-wide association studies (GWAS) using different facets of average methylation as the phenotype. For global CG and CHG methylation, no associations reached genome-wide significance, while for CHH methylation a clear peak of association was observed on chromosome 4 (*Figure 1—figure supplement 1*). The association was even more significant when restricting attention to average CHH methylation of large transposons (*Figure 1B*), in agreement with the notion that this type of methylation mostly occurs in transposons in *Arabidopsis* (*Stroud et al., 2013*).

The association centered around a SNP at 10,459,127 on chromosome 4, 38 kb downstream from the locus AT4G19020, which encodes a homolog of the CHG methyltransferase chromo-methylase-3 (*Lindroth et al., 2001*) that has recently been shown to catalyze both CHH and CHG methylation on transposons, and is thus an excellent candidate (*Zemach et al., 2013*; *Stroud et al., 2014*). A multi-locus mixed model (*Segura et al., 2012*) that included the identified SNP (*CMT2a*) as a fixed effect revealed another SNP downstream of *CMT2*, at position 10,454,628 (*CMT2b*), 4.5 kb closer to *CMT2* than *CMT2a*, and in complete linkage disequilibrium with it (i.e., the non-reference alleles at *CMT2a* and *CMT2b* are never seen together). Repeating the GWAS with both *CMT2a* and *CMT2b* as cofactors identified no further loci (*Figure 1—figure supplement 2*). Both non-reference alleles are common in southern Sweden, but are also found in the north (22.6% vs 9.5% and 30.6% vs 7.9% for *CMT2a* and *CMT2b*, respectively). Accessions with the non-reference *CMT2a* allele have on average more CHH methylation on transposons than those with the reference haplotype (p = 1.1e-03), while those with the non-reference *CMT2b* allele have lower levels of CHH methylation than the reference haplotype (p = 8.1e-03; *Figure 1C*). The associations were readily confirmed using an F2 population generated by

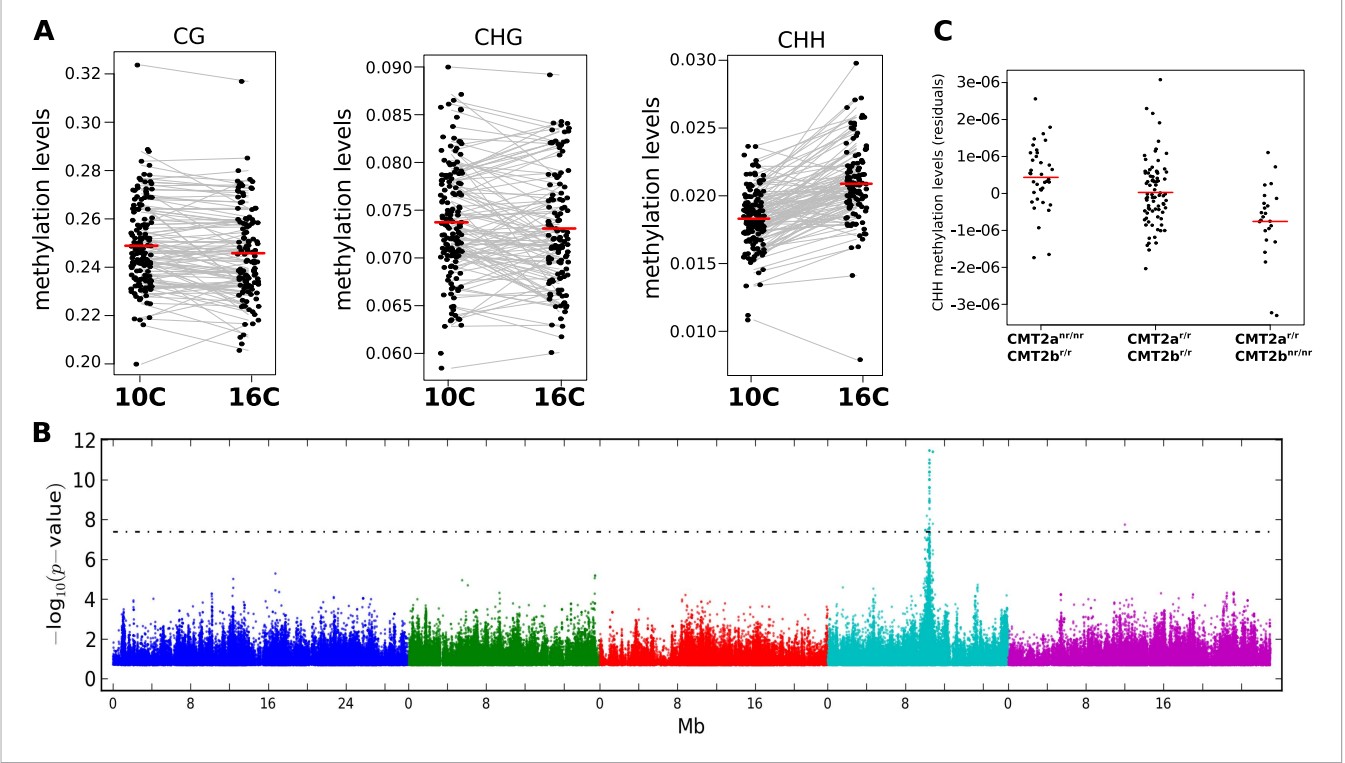

**Figure 1**. The effect of CMT2 on genome-wide CHH methylation levels. (**A**) Genome-wide average methylation level reaction norms for each accession (156 samples at 10˚C and 125 samples at 16˚C). Only CHH levels differ significantly between temperatures (Wilcoxon rank sum test; p = 1.7e-16). (**B**) Manhattan plot of genome-wide association studies (GWAS) results using average levels of CHH methylation for 151 accessions at 10˚C on large transposons as the phenotype (the peak is also seen at 16˚C [not shown]). The threshold line indicates a Bonferroni-corrected p-value of 0.05. (**C**) CHH methylation on large (over 2 kb) transposons at 10˚C by CMT2 two-locus genotype (population sizes are 36, 82, and 24 for CMT2a$^{nr/nr}$/CMT2b$^{r/r}$, CMT2a$^{r/r}$/CMT2b$^{r/r}$, CMT2a$^{r/r}$/CMT2b$^{nr/nr}$, respectively). The values plotted are the Best Linear Unbiased Predictor (BLUP) estimates after correcting for population structure. Since accessions are homozygous, only four genotypes are possible, of which only three exist due to complete linkage disequilibrium between *CMT2a* and *CMT2b*. **Figure 1—figure supplement 1** shows Manhattan plots of GWAS results for global methylation averages. **Figure 1—figure supplement 2** shows Stepwise GWAS using average CHH methylation of TE's.

The following figure supplements are available for figure 1:

**Figure supplement 1**. Manhattan plots of GWAS results for global methylation averages.

**Figure supplement 2**. Stepwise GWAS using average CHH methylation of TE's as a phenotype.

crossing accessions with the *CMT2a* and *CMT2b* non-reference alleles (**Figure 2**). No significant differences in CMT2 mRNA levels were observed between the alleles in our data and limited Sanger sequencing of cDNA showed no evidence of splicing variants (although, as will be discussed below, we did detect a putative rare null allele). Several non-synonymous polymorphisms in the methyltransferase and BAH domains of CMT2 were detected (**Supplementary files 1** and **2**) but they do not explain the phenotype as well as the *CMT2a* and *CMT2b* SNPs.

The effect of genetic variation on local CHH methylation was examined by calculating the methylation level in 200 bp sliding windows across the genome (100 bp overlap between windows) and running GWAS for the 200,000 differentially methylated regions (DMRs; see 'Materials and methods') that varied most between individuals. 36023 DMRs had at least one genome-wide significant association (Bonferroni-corrected p-value of 0.05; 7273 remain after correcting for multiple GWAS using an FDR of 0.05). 45% (15,031) of the DMRs had a significant *cis*-association within 100 kb, while the rest showed evidence of *trans*-regulation, including the dramatic effect of *CMT2* on chromosome 4 which accounted for approximately 21% (7392) of all significant associations (**Figure 3A**).

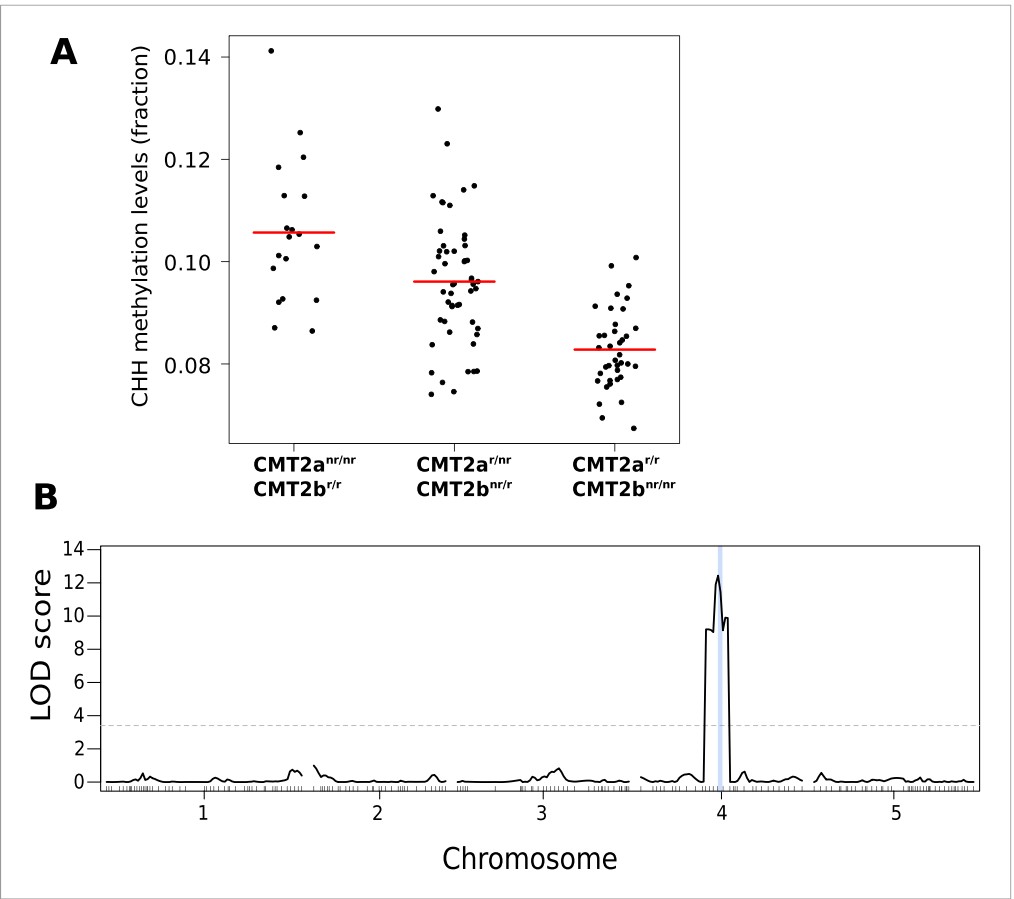

**Figure 2**. CHH methylation levels in an F2 population map to CMT2. (**A**) CHH methylation on large transposons by CMT2 genotype in an F2 population of 113 individuals (population sizes are 19, 52, and 38 for CMT2a$^{nr/nr}$/CMT2b$^{r/r}$, CMT2a$^{r/r}$/CMT2b$^{r/r}$, CMT2a$^{r/r}$/CMT2b$^{nr/nr}$, respectively; 4 individuals whose genotype at CMT2 could not be accurately inferred were omitted). (**B**) Mapping of CHH methylation of long TEs in the same population. The dotted line indicates a LOD threshold with a genome-wide p-value of 0.05 obtained using 1000 permutations, and the vertical blue line shows the marker interval that contains CMT2.

To quantify the regulation of DMRs, we partitioned the variance of CHH methylation into environmental (E), *CMT2*, *CMT2* X E, *cis*, *cis* X E, *trans*, and *trans* X E using a mixed model (*Figure 3B*). The analysis confirmed substantial *cis* and *trans* associations, with the environment modulating the genetic effects rather than having a major direct effect. At least for the *cis* associations, a possible explanation is that SNPs tag polymorphic TE insertions, with the insertion allele being silenced in a temperature-sensitive manner.

The effect of temperature on CHH methylation could also be seen at the local level. We defined 'temperature DMRs' by looking for windows that differed significantly between temperatures. Comparing 16°C–10°C, each accession on average gained CHH methylation at ~400 temperature DMRs and lost it at ~200 temperature DMRs (false discovery rate = 0.05). CHH methylation is associated with transposable elements (TEs; *Finnegan et al., 1998*), and in agreement with this, 79% of the temperature DMRs where methylation was gained at 16°C were located within 500 bp of an annotated TE (with 60% directly overlapping one). These temperature DMRs were enriched in a small subset of TEs (835, or 2.7% of total, permutation based p-value = 0.05) that were more highly methylated than other transposons, but with lower methylation levels immediately adjacent (*Figure 4A*). Compared to TEs without temperature DMRs, these 'variable' TEs also tended to be euchromatic (*Figure 4B*), highly expressed (*Figure 4C*), and recently inserted ('evolutionarily young' TE insertions for which orthologs are not present in *Arabidopsis lyrata* [*Zhong et al., 2012*] comprised 75% of the variable TEs vs 68% of non-variable TEs). At the super-family level, members

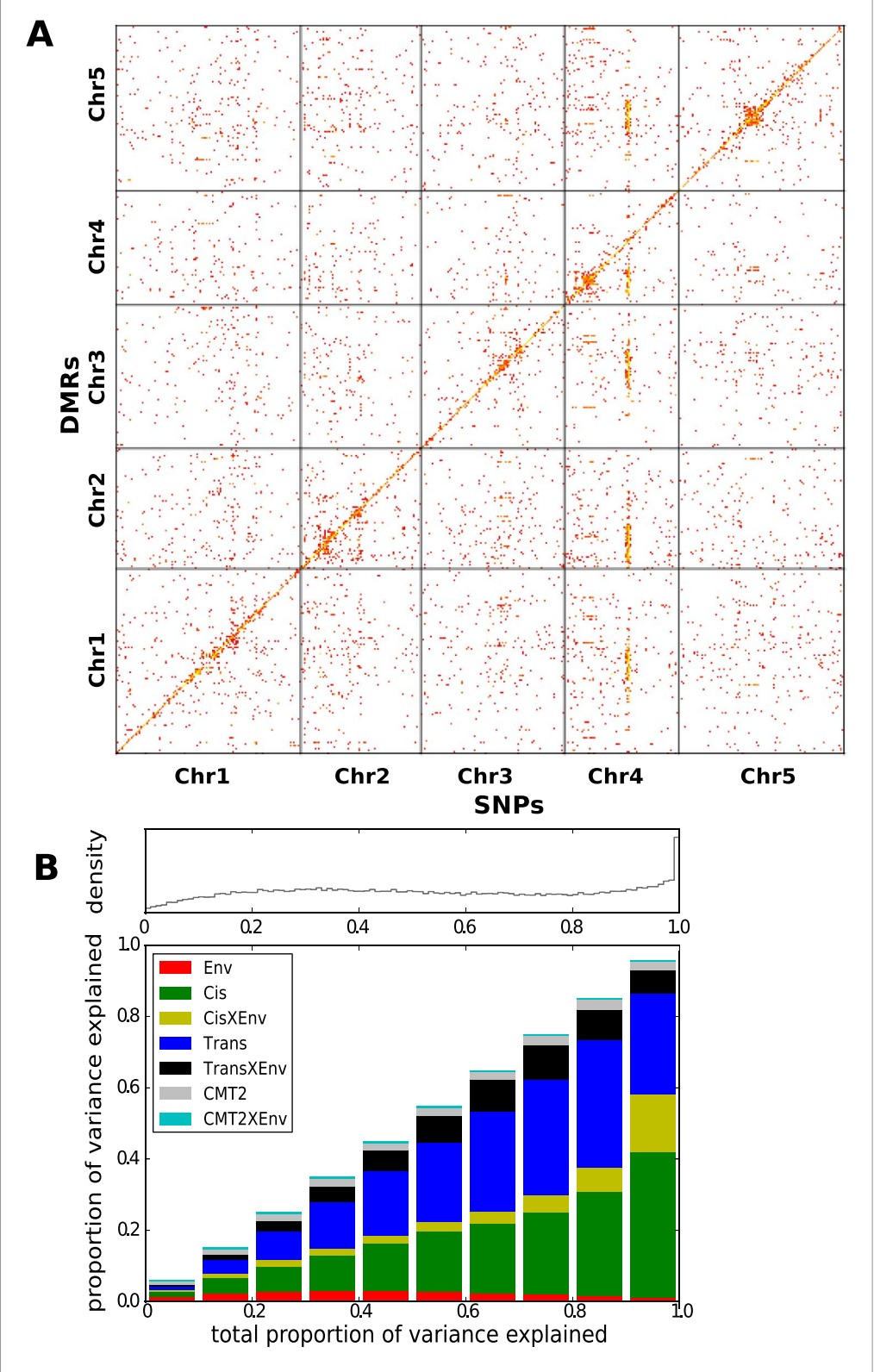

**Figure 3**. Genetic basis CHH methylation variation. (**A**) GWAS for CHH differentially methylated regions (DMRs) at 10°C in 151 accessions, defined using 200 bp sliding windows across the genome and selecting the 200,000 most variable ones. For each DMR, SNPs significantly associated at the Bonferroni-corrected 0.05-level are plotted. (**B**) Variance-components analysis of the CHH DMRs. For each DMR, a mixed model with *cis*, *CMT2*, and

*Figure 3. Continued*

genome-wide *trans* effects, plus environment and genetic interactions with environment was fitted (see 'Materials and methods'). DMRs were binned by the total variance explained by the model. The density of DMRs in each bin is shown at the top, and the bottom shows the average variance-decomposition for each bin.

of the *SINE*, *SINE*-like, *Helitrons* and *Mutator*-like DNA TE superfamilies were over-represented among the variable transposons, and at the family level, 36 families were over-represented, including the *AtREP*, *Vandal* and *HAT* DNA transposons, as well as *COPIA78/ONSEN* and *META1* retroelements (*Table 1*). Interestingly, *COPIA78* has been shown to become active in response to heat stress (*Pecinka et al., 2010*; *Ito et al., 2011*) apparently due to heat-shock promoter elements in its LTR regions (*Cavrak et al., 2014*).

In order to gain further insight into the mechanisms responsible for variation in CHH methylation, we bisulfite-sequenced knockout lines of CMT2 (SAIL_906_G03) and DCL3 (*dcl3-5* [*Daxinger et al., 2009*], a component of the RdDM pathway), and identified 10,138 DCL3-dependant DMRs and 33,422 CMT2-dependent DMRs as described in section 'DMR calling on DNA methylation mutants' of the 'Materials and methods'. As expected under the assumption that CMT2 is responsible for the massive GWAS peak on chromosome 4, the GWAS peak at this locus remains if we consider only the CMT2-dependent DMRs, but not for DCL3-dependent DMRs (*Figure 5—figure supplement 1*). Furthermore, while CHH methylation varied with temperature at both DCL3- and CMT2-dependent DMRs (*Figure 5*), DCL3-dependent DMRs were much more strongly associated with previously identified temperature DMRs (4703 out of 10,138 DCL3-dependent DMRs, or 46%, overlapped temperature-sensitive DMRs, whereas the corresponding numbers for CMT2-dependent DMRs were 2299 out of 33,422, or 7%; Fisher's exact p-value < 2.2e-16), suggesting that much of the temperature variation in CHH methylation is due to components of the RdDM pathway. This result is consistent with previous findings showing that RNA silencing is less active at lower temperatures (*Romon et al., 2013*).

Interestingly, we observed one accession from northern Sweden, TAA-03, with almost undetectable levels of CHH methylation at CMT2-dependant DMRs (*Figure 5*). Further investigation suggested that it has a deletion or rearrangement in CMT2, as we were unable to map reads between positions 2813 and 4944 (intron 7 to exon 16, *Figure 5—figure supplement 2*). Sanger-sequencing indicates the insertion of a stretch of TC dinucleotide repeats of at least 330 bp. The same deletion appears to be present in three more accessions from northern Sweden (TAA-14, TAA-18, and Gro-3) a situation reminiscent of the homologous CMT1 gene, which seems to be non-functional in most *Arabidopsis* accessions (*Henikoff and Comai, 1998*). Although CMT2 null alleles have no obvious phenotype, the gene is highly conserved in plants (with the exception of maize; *Zemach et al., 2013*; *West et al., 2014*).

It has recently been suggested that natural variation in CMT2 is associated with adaptation to climate (*Shen et al., 2014*), although the alleles identified in that study do not overlap with the ones identified here. Given the sensitivity of CHH methylation to growth temperature observed here, we next investigated the correlation between DNA methylation and the climate of origin (*Hancock et al., 2011*). While CHH methylation was moderately correlated with photosynthetically active radiation (PAR) in spring (Pearson's r = 0.38), and CHG showed correlation with aridity (r = 0.35) and the number of frost-free days (Pearson's r = 0.30), by far the strongest signal was a strong positive correlation between CG methylation and latitude (Pearson's r = 0.70), as well as with a number of environmental variables that co-vary with latitude in our sample, such as minimum temperature and daytime length (*Table 2*, *Figure 6A*). As a result of the strong latitudinal correlation, accessions originating from northern Sweden (minimum temperature below −10°C) had on average 11% higher global CG methylation compared to those from the south (*Figure 6A*). The correlation appears to be driven by gene body methylation (GBM): as the correlation for CG methylation on transposons was much weaker (*Figure 6A*, *Figure 6—figure supplement 1*). Because the methylation variation observed for genes with average CG methylation below 5% appeared mostly to be noise (*Figure 6—figure supplement 2*, see also the 'Materials and methods' section), we classified genes into 'unmethylated' and 'having GBM' using this as a cutoff (5%). We also eliminated genes showing a transposon-like pattern of methylation in which not only CG, but also the CHH and CHG contexts are highly methylated (*Zemach et al., 2013*). In what follows, we use GBM to refer only to

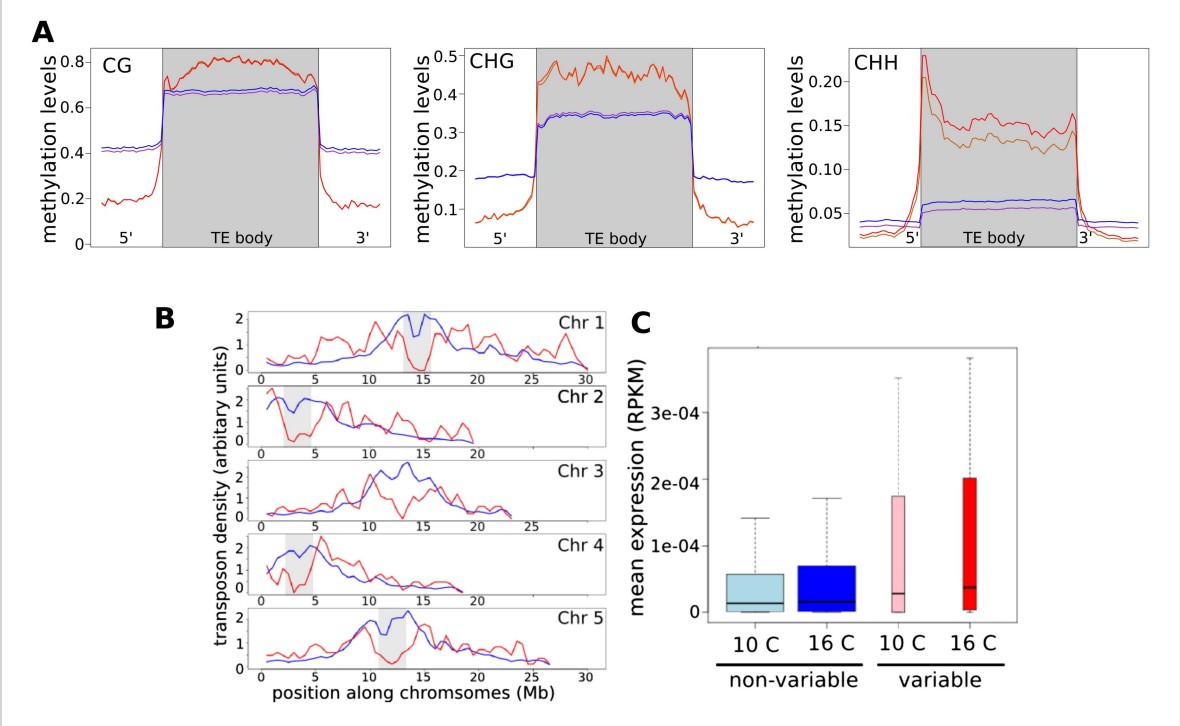

**Figure 4**. CHH methylation varies with temperature. (**A**) Average methylation levels over variable transposons at 10°C (orange) vs 16°C (red), and over non-variable transposons at 10°C (purple) vs 16°C (dark blue). Methylation for variable TEs is significantly higher (permutation p-value for CHH methylation = 0.05). (**B**) The density of variable (red) and non-variable (blue) TEs along chromosomes in 500 kb windows. Density is defined as the percentage of the total number in either category in each window; pericentromeric regions are shaded grey. (**C**) The expression of TEs at both temperatures. Variable TEs are more highly expressed than non-variable TEs, but the difference is only statistically significant at 16°C (Wilcoxon: 10°C, p = 0.15; 16°C, p = 0.023).

gene body CG methylation for this filtered set. In order to better understand the observed variation in GBM, we examined CG methylation at the single nucleotide resolution within GBM containing genes. Although methylation was detectable (using a cut-off of 1%) at a similar number of sites in the north and south (1085292 vs 1079443 CG sites), those in the north showed a distinct skew towards higher methylation levels (*Figure 6—figure supplement 3*). Likewise when the difference between average methylation levels in the north and south was calculated individually for each of the cytosines, the majority of cytosines showed a small increase in the north compared to the south (*Figure 6—figure supplement 4*). From this we concluded that there is a general small increase in methylation of most CG dinucleotides in GBM genes, rather than large changes in a specific subset. GBM primarily occurs on long, evolutionary conserved genes that tend to be moderately-to-highly expressed, and is positively correlated with gene expression (*Zilberman et al., 2007*; *Takuno and Gaut, 2012*). Genes with higher GBM tended to be more highly expressed in our data as well, and—more interestingly—accessions with higher average GBM showed slightly higher average expression of methylated genes (although the correlation was weak, *Figure 6—figure supplement 5*). Given that northern accessions had higher GBM, this meant that genes with GBM were on average more highly expressed in northern than in southern accessions, while unmethylated genes showed little difference (*Figure 6B*). GBM has previously been shown to be anti-correlated with temperature-dependent gene expression (*Kumar and Wigge, 2010*). While no large-scale north-south expression differences were observed between 10°C and 16°C in our data, northern accessions showed considerably less variation in expression between the two temperatures for genes with GBM (Wilcoxon p-value = 1.2e-05), while no such difference was observed for genes without it (*Figure 6—figure supplement 6*).

As for CHH DMRs, the genetic basis of GBM was examined using a variance-component approach (*Figure 7A*). The results were dramatically different: relative to CHH methylation, the *trans* effects for GBM are massive, and the environment appears to have no effect (in agreement with the observation

**Table 1**. Super-families (italics) and families that are over-represented among 'variable' TEs

| TE (*super-*)family | Expected | Observed | Enrichment | 95th Quantile |
|---|---|---|---|---|
| *RathE1_cons* | *5* | *26* | *4.56* | *10* |
| *RathE3_cons* | *2* | *9* | *3.23* | *6* |
| *RathE2_cons* | *1* | *5* | *2.52* | *4* |
| *SINE* | *3* | *7* | *2.00* | *7* |
| *RC/Helitron* | *346* | *444* | *1.28* | *368* |
| *DNA/MuDR* | *144* | *184* | *1.27* | *162* |
| ATREP2 | 4 | 53 | 12.07 | 8 |
| RP1_AT | 2 | 27 | 11.59 | 5 |
| ATTIRX1C | 1 | 12 | 11.49 | 3 |
| ATREP13 | 2 | 28 | 9.87 | 6 |
| VANDAL22 | 1 | 11 | 8.56 | 3 |
| SIMPLEHAT1 | 1 | 11 | 7.34 | 4 |
| VANDAL2N1 | 1 | 10 | 7.32 | 3 |
| ATREP8 | 2 | 13 | 6.47 | 5 |
| VANDAL2 | 1 | 7 | 6.08 | 3 |
| ATREP10 | 1 | 10 | 5.93 | 4 |
| AT9NMU1 | 1 | 7 | 5.81 | 3 |
| ATN9_1 | 1 | 10 | 5.75 | 4 |
| SIMPLEHAT2 | 1 | 11 | 5.63 | 4 |
| META1 | 3 | 20 | 5.41 | 7 |
| ATDNAI27T9A | 3 | 15 | 4.83 | 6 |
| ATREP2A | 3 | 15 | 4.83 | 6 |
| ATCOPIA78 | 0 | 3 | 4.67 | 2 |
| VANDAL18NA | 0 | 3 | 4.67 | 2 |
| RathE1_cons | 5 | 26 | 4.56 | 10 |
| VANDAL14 | 0 | 3 | 4.48 | 2 |
| SIMPLEGUY1 | 3 | 13 | 4.19 | 6 |
| ATDNATA1 | 0 | 3 | 4.00 | 2 |
| TNAT2A | 1 | 4 | 3.93 | 3 |
| ATREP7 | 4 | 16 | 3.64 | 8 |
| RathE3_cons | 2 | 9 | 3.23 | 6 |
| ATREP14 | 1 | 4 | 3.18 | 3 |
| ATREP16 | 1 | 4 | 3.18 | 3 |
| LIMPET1 | 3 | 9 | 2.90 | 6 |
| ATREP6 | 4 | 14 | 2.89 | 8 |
| ARNOLDY2 | 7 | 22 | 2.85 | 13 |
| ATSINE4 | 2 | 7 | 2.67 | 5 |
| ATDNAI27T9C | 2 | 7 | 2.42 | 6 |
| ATREP3 | 38 | 92 | 2.39 | 49 |
| ARNOLDY1 | 6 | 14 | 2.21 | 11 |
| ATREP1 | 13 | 23 | 1.73 | 19 |
| HELITRONY3 | 37 | 51 | 1.36 | 48 |

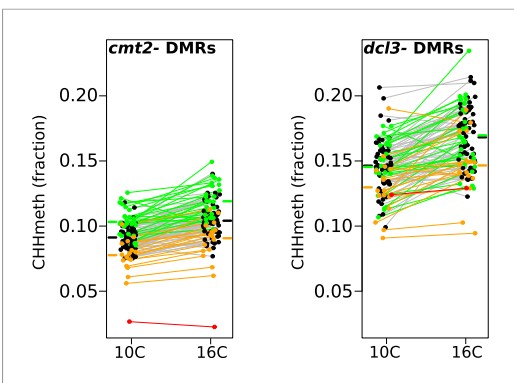

**Figure 5**. Temperature dependent CHH methylation variation at RdDM and CMT2 controlled DMRs. CHH methylation at CMT2- and DCL3-dependent DMRs in natural accessions grown at 10°C and 16°C (cf. *Figure 1A*, each population has 110 individuals). The difference between temperatures was highly significant for both types of DMR (Wilcoxon p-value = 9.1e-11 and p-value = 5.9e-12 respectively). Black points/grey lines indicate accessions with the *CMT2* reference allele; green, the *CMT2a* non-reference allele; and orange, the *CMT2b* non-reference allele. Red is the TAA-03 accession, which has a putative null allele of CMT2. Average methylation levels for each of the genotypes are shown in bars to the side *Figure 5—figure supplement 1* shows GWAS on CMT2 and DCL3 dependant DMRs. *Figure 5—figure supplement 2* shows a putative null allele of CMT2.

The following figure supplements are available for figure 5:

**Figure supplement 1**. GWAS on CMT2 and DCL3 dependent DMRs.

**Figure supplement 2**. Putative null allele of CMT2.

that only CHH methylation levels vary with temperature, see *Figure 1A*). To identify the genes responsible, we also performed GWAS for each gene with GBM (*Figure 7B*). A total of 3241 significant associations were found for 2315 genes. 43% of these genes had a significant *cis*-association (within 100 kb of the gene of interest), which could represent local variants affecting methylation directly, or indirectly by affecting gene expression (*Gutierrez-Arcelus et al., 2013*). No evidence for major *trans*-acting loci like *CMT2* was found, but 69% of all significant associations were in *trans*. A comparison of the direction of the effect of GBM-associated SNPs in *cis* and *trans* revealed a striking pattern (*Figure 7C*). While the non-reference alleles of *cis*-SNPs were 1.18 times more likely to be associated with decreased rather than increased GBM (p = 2.01e-04), the non-reference alleles of *trans*-SNPs were 3.48 times more likely to be associated with increased GBM (p = 2.2e-16), and the non-reference alleles at the 15 *trans*-SNPs that were associated with GBM at five or more genes were always positively correlated (*Figure 7C*). Furthermore, while *cis*-SNPs showed a wide distribution of allele frequencies similar to random SNPs, *trans*-SNPs showed a much more limited distribution of frequencies (*Figure 8A*) and were also much more strongly correlated with latitude (*Figure 8B,C*). The correlation between GBM and latitude thus appears mostly to be due to *trans*-acting SNPs.

The 15 highly associated *trans*-SNPs were largely limited to northern Sweden, and were in strong linkage disequilibrium with each other (*Figure 8—figure supplement 1*). *A. thaliana* from northern Sweden show signs of multiple strong selective sweeps (*Long et al., 2013*) and harbors many polymorphisms that appear to be involved in local adaptation (specifically to minimum temperature; *Hancock et al., 2011*). The 15 SNPs were more than ninefold over-represented in the previously identified sweep regions (empirical p-value = 5.1e-03) and over fivefold over-represented within 2 kb of SNPs in the 1% tail of those associated with minimum temperature (*Hancock et al., 2011*) (empirical p-value = 3.1e-04), (*Table 3*). The ancestral state could be determined for 10 of the 15 SNPs, and in 8 of these cases, the non-reference allele was derived, further supporting sweeps in northern Sweden.

That the difference in GBM between north and south is likely to reflect local adaptation is also clear from its relative magnitude. The north vs south divide explains a much higher fraction of the additive genetic variance for GBM (Qst = 0.772; see 'Materials and methods') than of the SNP variance (Fst = 0.187). This strongly suggest that the phenotypic differentiation is driven by selection rather than genetic drift (*Leinonen et al., 2013*).

Identifying the causal variants is challenging, a gene-ontology analysis of genes within 100 kb (the average size of the sweep regions, *Long et al., 2013*), of the 15 *trans*-SNPs found enrichment of loci associated with mRNA transcription (GO0009299, p-value = 2.62e-03). Genes involved in epigenetic processes are not captured well by standard gene-ontology, but we found that genes from the plant chromatin database (www.chromdb.org/) were significantly overrepresented in these regions as well (permutation p-value = 0.012; *Table 4*).

**Table 2.** Correlation between methylation levels and environment-of-origin variables (*Hancock et al., 2011*)

| Environmental variable | Growing temp. | CG | | | CHG | | | CHH | | |
|---|---|---|---|---|---|---|---|---|---|---|
| | | r | rho | p-value | r | rho | p-value | r | rho | p-value |
| Latitude | 10 | 0.69 | 0.52 | 7.8E-11 | −0.24 | −0.19 | 2.7E-02 | 0.10 | 0.14 | 1.1E-01 |
| | 16 | 0.62 | 0.47 | 3.2E-07 | −0.21 | −0.20 | 4.2E-02 | 0.04 | −0.11 | 2.5E-01 |
| Longitude | 10 | 0.59 | 0.54 | 1.2E-11 | −0.14 | −0.09 | 3.1E-01 | 0.23 | 0.28 | 7.5E-04 |
| | 16 | 0.55 | 0.53 | 4.4E-09 | −0.12 | −0.03 | 7.4E-01 | 0.14 | 0.15 | 1.2E-01 |
| Temperature seasonality | 10 | 0.68 | 0.49 | 1.6E-09 | −0.27 | −0.24 | 4.8E-03 | 0.09 | 0.09 | 2.8E-01 |
| | 16 | 0.62 | 0.42 | 1.1E-05 | −0.23 | −0.26 | 6.6E-03 | 0.04 | −0.12 | 2.1E-01 |
| Max. temp. (warmest month) | 10 | −0.14 | 0.06 | 4.6E-01 | −0.07 | −0.13 | 1.3E-01 | 0.14 | 0.20 | 2.0E-02 |
| | 16 | −0.03 | 0.10 | 2.9E-01 | −0.10 | −0.20 | 3.8E-02 | 0.05 | 0.03 | 7.3E-01 |
| Min. temp. (coldest month) | 10 | −0.70 | −0.56 | 9.1E-13 | 0.27 | 0.21 | 1.2E-02 | −0.07 | −0.06 | 4.7E-01 |
| | 16 | −0.63 | −0.48 | 2.7E-07 | 0.24 | 0.24 | 1.4E-02 | 0.00 | 0.19 | 5.6E-02 |
| Precipitation (wettest month) | 10 | 0.45 | 0.52 | 1.2E-10 | −0.25 | −0.27 | 1.2E-03 | −0.20 | −0.12 | 1.7E-01 |
| | 16 | 0.29 | 0.43 | 4.0E-06 | −0.26 | −0.24 | 1.2E-02 | −0.22 | −0.19 | 5.8E-02 |
| Precipitation (driest month) | 10 | 0.31 | 0.40 | 1.5E-06 | −0.33 | −0.29 | 6.5E-04 | −0.24 | −0.21 | 1.6E-02 |
| | 16 | 0.21 | 0.32 | 7.4E-04 | −0.26 | −0.24 | 1.4E-02 | −0.15 | −0.18 | 6.0E-02 |
| Precipitation seasonality | 10 | 0.42 | 0.44 | 7.1E-08 | −0.07 | −0.16 | 5.4E-02 | 0.05 | 0.01 | 9.0E-01 |
| | 16 | 0.36 | 0.37 | 1.2E-04 | −0.13 | −0.16 | 1.1E-01 | 0.01 | −0.01 | 9.1E-01 |
| PAR (spring) | 10 | 0.04 | 0.22 | 8.9E-03 | 0.20 | 0.18 | 3.7E-02 | 0.24 | 0.23 | 7.3E-03 |
| | 16 | 0.03 | 0.18 | 6.6E-02 | 0.27 | 0.21 | 3.5E-02 | 0.38 | 0.35 | 2.8E-04 |
| Length of growing season | 10 | −0.59 | −0.57 | 5.5E-13 | 0.24 | 0.23 | 7.3E-03 | −0.16 | −0.18 | 3.3E-02 |
| | 16 | −0.58 | −0.54 | 4.0E-09 | 0.23 | 0.21 | 3.0E-02 | −0.04 | 0.01 | 8.9E-01 |
| No. consecutive cold days | 10 | 0.60 | 0.53 | 4.0E-11 | −0.19 | −0.13 | 1.2E-01 | 0.17 | 0.28 | 1.1E-03 |
| | 16 | 0.57 | 0.53 | 4.2E-09 | −0.17 | −0.09 | 3.7E-01 | 0.10 | 0.08 | 4.1E-01 |
| No. consecutive frost-free days | 10 | −0.59 | −0.49 | 1.2E-09 | 0.29 | 0.27 | 1.5E-03 | 0.02 | 0.03 | 7.1E-01 |
| | 16 | −0.51 | −0.39 | 4.9E-05 | 0.30 | 0.30 | 1.6E-03 | 0.07 | 0.13 | 1.9E-01 |
| Relative humidity (spring) | 10 | 0.62 | 0.47 | 5.6E-09 | −0.23 | −0.18 | 3.9E-02 | 0.09 | 0.06 | 4.5E-01 |
| | 16 | 0.53 | 0.37 | 1.2E-04 | −0.20 | −0.26 | 7.6E-03 | 0.04 | −0.08 | 4.3E-01 |
| Daylength (spring) | 10 | 0.69 | 0.50 | 7.2E-10 | −0.27 | −0.21 | 1.4E-02 | 0.08 | 0.05 | 5.7E-01 |
| | 16 | 0.63 | 0.41 | 1.5E-05 | −0.23 | −0.29 | 2.7E-03 | 0.04 | −0.17 | 8.7E-02 |
| Aridity | 10 | 0.53 | 0.49 | 8.4E-10 | −0.35 | −0.31 | 1.9E-04 | −0.18 | −0.21 | 1.3E-02 |
| | 16 | 0.43 | 0.42 | 8.4E-06 | −0.28 | −0.24 | 1.3E-02 | −0.13 | −0.20 | 3.8E-02 |

r = Pearson's correlation, rho = Spearman's rank correlation, p-value = significance of rho.

PAR = photosynthetically active radiation.

We also looked for genes whose expression variation is consistent with a causal role. We identified 68 genes within 100 kB of one of the 15-*trans* SNPs whose expression is highly correlated (Wilcoxon test p < 0.001) with the adjacent SNP after correction for population structure (*Table 5*). No significant enrichment of Gene Ontology terms was observed among these genes, and manual inspection identified no proteins directly involved in DNA methylation. Instead, a number of proteins involved in the regulation of gene expression and/or chromatin accessibility were present (*Table 5*). This may suggest that the increased gene-body methylation observed in the north is not directly due to increased DNA methylation, but may be caused by increases in gene expression driven either by differences in transcription factors networks or chromatin compaction. Identification of the causal variants behind this phenomenon should provide insight into how plants adapt to their local environment.

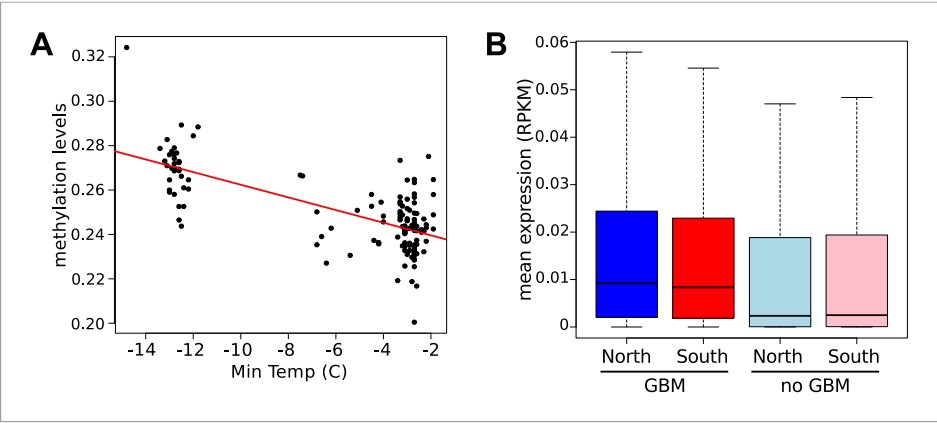

**Figure 6**. Latitudinal difference in gene body methylation (GBM) and gene expression. (**A**) Global CG methylation levels at 10°C for 151 accessions are strongly correlated with minimum temperature at the location of origin. Results for 16°C are similar. (**B**) Genes with GBM are more highly expressed at 10°C in northern (blue) than in southern (red) accessions (wilcoxon rank sum test p = 2.1e-03), whereas genes without GBM show little difference (p = 1.9e-02). At 16°C the difference for genes with GBM is more significant (p = 6.4e-05), whereas the difference for genes without GBM is insignificant (p = 0.49).

The following figure supplements are available for figure 6:

**Figure supplement 1**. Correlation between CG methylation levels and the minimum temperature at location of origin.

**Figure supplement 2**. Filtering of GBM variation data.

**Figure supplement 3**. Distribution of methylation levels at individual CG dinucleotides within GBM genes.

**Figure supplement 4**. Distribution of variation in methylation levels between the north and the south for individual CG dinucleotides within GBM genes.

**Figure supplement 5**. Accessions with higher average GBM tend to have higher average expression (of genes with GBM, normalized by genes without GBM; r = 0.131, p = 0.0386).

**Figure supplement 6**. Genes with GBM show less expression variation between temperatures.

In conclusion, genes with GBM are generally up-regulated and more heavily methylated in northern accessions, and the change appears to be due to *trans*-acting polymorphisms that have been subject to directional selection. The candidate regions show an overrepresentation of genes involved in transcriptional processes. We also found that CHH methylation of TEs is temperature sensitive, and identified a major *trans*-acting controller, *CMT2*. Taken together, these observations suggest that local adaptation in *A. thaliana* involves genome-wide changes in fundamental mechanisms of gene regulation, perhaps as a form of temperature compensation.

## Materials and methods

### Raw data generation

#### Plant growth
A diverse set of 150 Swedish accessions were sown on soil and stratified for 3 days at 4°C in the dark. They were then transferred to environmentally controlled growth chambers set at 10°C or 16°C under long day conditions (04:00–20:00) and individual seedlings were transplanted to single pots after 1 week. When plants attained the 9-true-leaf stage of development, whole rosettes were collected between 15:00 and 16:00 hr and flash frozen in liquid nitrogen.

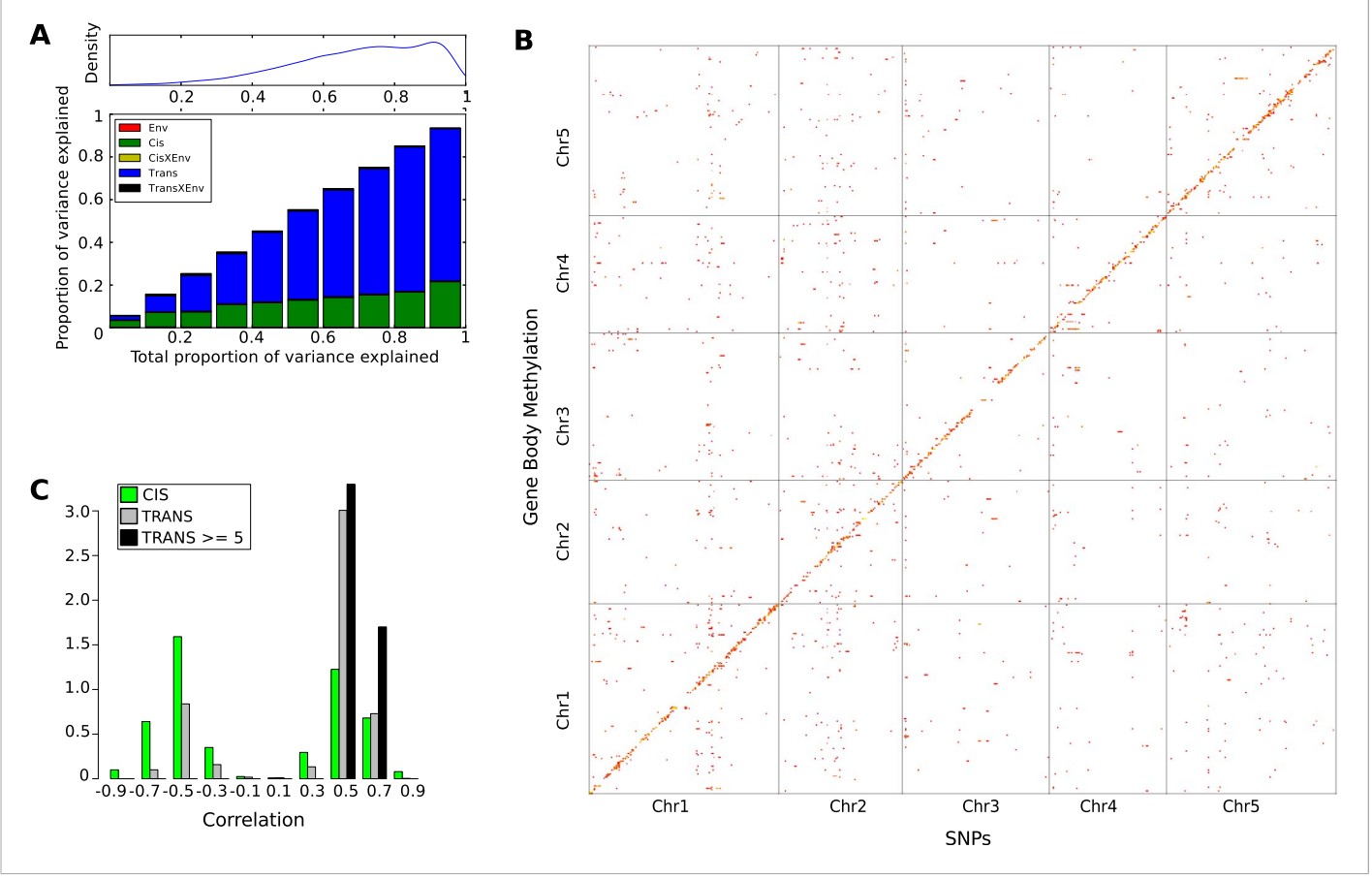

**Figure 7**. The genetic basis of GBM. (**A**) Variance component analysis of GBM. (**B**) Significant associations (Bonferroni-corrected 0.05-level) from a GWAS of GBM for individual genes. (**C**) Correlation between non-reference allele at associated SNPs and GBM.

In addition, a cross between the T550 and Brösarp-11-135 accessions was created and F2 seed obtained. 113 individual F2 lines were grown in the same manner as the accessions.

## RNA-seq library preparation

For each accession, three plants were pooled and total RNA was extracted by TRIzol (Invitrogen, Carlsbad, California, 15596-018), DNase treated and mRNA purified with oligo dT Dynabeads (Life Technologies, Carlsbad, California). RNA was then fragmented using Ambion Fragmentation buffer (Life Technologies) and first and second strand cDNA synthesis was carried out using Invitrogen kit 18064-071. The ends of sheared fragments were repaired using Epicentre (Madison, Wisconsion) kit ER81050. After A-tailing using exo-Klenow fragment (New England Biolabs, Ipswich, Massachusetts, NEB M0212L), barcoded adaptors were ligated with Epicentre Fast-Link DNA Ligation Kit (Epicentre LK6201H). Adaptor-ligated DNA was resolved on 1.5% low melt agarose gels for 1 hr at 100 V. DNA in the range of 200–250 bp was excised from the gel and purified with the Zymoclean Gel DNA recovery kit (Zymo Research). The libraries were amplified by PCR for 15 cycles with Illumina PCR primers 1.1 and 1.2 with Phusion polymerase (NEB F-530L).

Single-end 32 bp sequencing was performed at the University of Southern California Epigenome Center on an Illumina (San Diego, California) GAIIx instrument using fourfold multiplexing.

## MethylC-seq library preparation

For each accession two individual plants were pooled and total DNA was extracted using CTAB and phenol-chloroform. Approximately two micrograms of genomic DNA was used for MethylC-seq library construction and sequencing (92 bp paired-end) by BGI.

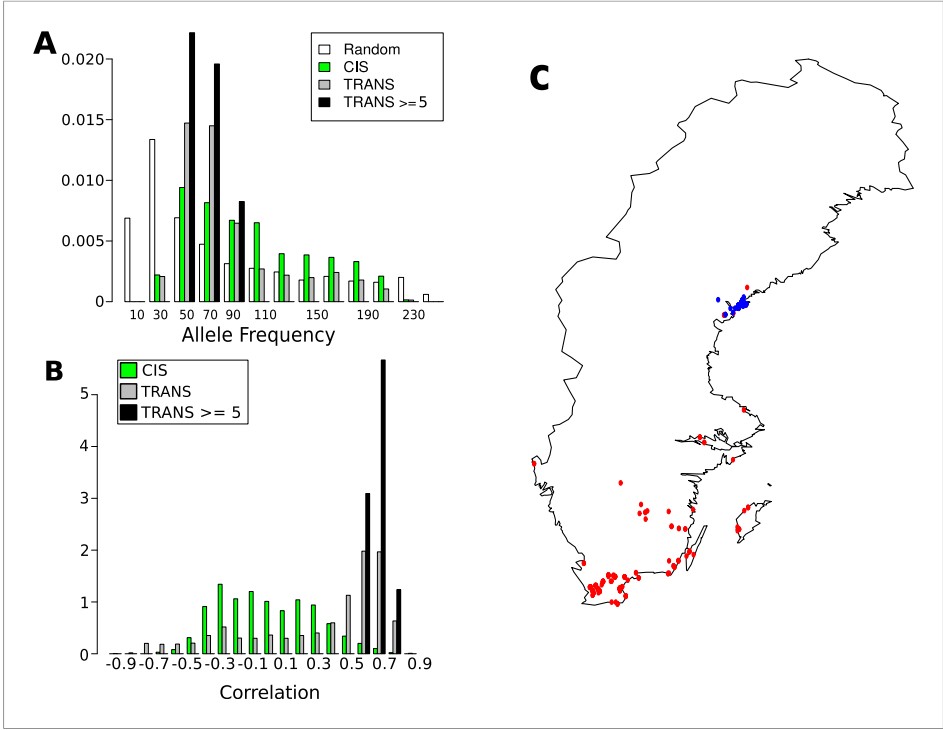

**Figure 8**. Frequency and distribution of GBM associated SNPs. (**A**) Correlation between non-reference allele at associated SNPs and latitude. (**B**) Non-reference allele frequency distribution for *cis* and *trans* SNPs compared to random SNPs. (**C**) Accessions carrying the non-reference alleles are limited to northern Sweden (accessions with the non-reference allele at 8 or more of the 15 loci blue, remaining accessions are red).

The following figure supplement is available for figure 8:

**Figure supplement 1**. Linkage disequilibrium between the 15 highly associated trans-SNPs.

## Sequence analysis

### Genome sequences

Illumina sequencing data from 180 published Swedish genomes (*Long et al., 2013*) were combined with sequencing data from another 79 (1001genomes.org), which had been processed using the same pipeline to yield polymorphism data for a total of 259 accessions (including those used for MethylC-seq and RNA-seq here). Linkage disequilibrium calculated using the R package LDHeatmap (version 0.9.1; *Shin et al., 2006*).

### RNA-seq data processing

#### Read mapping

After demultiplexing, 36 bp RNA-Seq reads were trimmed from barcodes (4 nt) and mapped to the TAIR10 reference genome including known variation with the PALMapper aligner (*Jean et al., 2010*) using a variant-aware alignment strategy. Two different sources of variants were used: (1) single nucleotide variants (SNV) and structural variants (SV) from genome sequencing (2.1) and (2) SNVs and SVs called in an initial alignment round of the RNA-Seq reads to the TAIR10 reference genome with PALMapper (relevant parameters: -M 4 -G 4 -E 6 -I 25000 -NI 1 -S). For both sources of variants we applied stringent filter criteria to reduce false calls: (1) genome variants had to appear in at least 40 strains with a minor allele count of at least 5 strains, (2) RNA-Seq variants had to be confirmed by at least 2 alignments within the same strain and had to have less than factor 2 many non-confirming alignments within the same strain. Variants from both sources were integrated into one file that was used for a second, variant-aware alignment round with

**Table 3**. 15 SNPs associated with gene body methylation (GBM) at 5 or more genes

| Chr | Position | Associated with GBM at how many genes? | Non-reference allele count | SNP-latitude correlation | Overlap with sweep (*Long et al., 2013*) | Overlap with min. temp. Assoc. SNPs (*Hancock et al., 2011*) |
|---|---|---|---|---|---|---|
| 1 | 912291 | 8 | 42 | 0.73 | none | 1_914088_0.21 |
| 1 | 4405103 | 5 | 66 | 0.64 | none | none |
| 1 | 7614101 | 5 | 48 | 0.66 | none | none |
| 1 | 19755967 | 5 | 88 | 0.75 | none | 1_19757140_0.24 |
| 2 | 6998631 | 6 | 55 | 0.87 | 2_6931030 | none |
| 2 | 7655016 | 6 | 81 | 0.61 | 2_7613651 | none |
| 2 | 7660469 | 9 | 55 | 0.78 | 2_7613651 | 2_7662427_0.30 |
| 2 | 7666059 | 5 | 69 | 0.72 | 2_7613651 | 2_7665507_0.25 |
| 2 | 7680882 | 5 | 82 | 0.63 | 2_7613651 | none |
| 2 | 7915712 | 6 | 51 | 0.83 | none | 2_7913782_0.23 |
| 2 | 9382495 | 5 | 73 | 0.71 | none | 2_9383856_0.34 |
| 2 | 9653878 | 9 | 48 | 0.80 | none | none |
| 3 | 419309 | 8 | 66 | 0.68 | none | none |
| 4 | 519982 | 8 | 66 | 0.70 | none | none |
| 4 | 13290034 | 5 | 74 | 0.74 | none | none |

PALMapper (relevant parameters: -M 2 -G 0 -E 2 -I 5000 -NI 0 -S). In variant-aware alignment mode, PALMapper builds an implicit representation of the reference genome that reflects all possible variant combinations that exist for a genomic region. The output is automatically projected to the TAIR10 coordinate system. To account for reads ambiguously mapping to multiple locations in the genome, we used a custom python script (*Supplementary file 3*) to remove all reads that showed at least one mapping additional to the best hit with the same edit distance. Additional hits were only counted as ambiguous, if they differed at least 3 nt in start and stop coordinates to the best hit. On average, 5.7 M reads were mapped per sample after removal of ambiguous reads. Low complexity libraries with less than 30% of mappable reads or samples with less than 800,000 mappable reads (6 in total) where excluded from further analysis.

## Sample validation

To correct for possible sample or data mix-ups, SNP were called from the RNA-seq alignments using a custom python script and compared to independently collected SNPs from the *Arabidopsis* 250K SNP array (Supplementary materials; *Kim et al., 2007*). Samples not matching the expected genotype were reassigned to the correct genotype where possible or otherwise excluded from further analysis.

## Filter for gene expression quantification

We quantified gene expression by counting the number of reads that were longer than 24 bp and that mapped to genes on all non-chloroplast and non-mitochondrial chromosomes. To obtain a stable quantification, we only used those reads that were uniquely mapped into the exonic regions of genes. Furthermore, we required that the reads did not map completely into regions where two genes overlap in order to avoid mixing quantifications of different genes. In the later text we will refer to this estimate as the raw expression estimate.

We also quantified the gene expression when additionally accounting for SV, alternative splicing and repetitive sequences that can all bias gene expression quantification. This estimate will be referred to as sv-corrected expression. For this quantification we additionally filtered for reads that start in an insertion or deletions and their two neighboring bases, that mapped into regions that are not contained in all transcripts of a gene and reads which were mapped completely into regions which are repetitive based on a 50 bp window.

**Table 4**. Genes in the plant chromatin database that are within 100 kb of one of the 15 SNPs associated with GBM at 5 or more genes

| ChromDB | Locus |
|---------|-------|
| ARID3 | AT2G17410 |
| ARP3 | AT1G13180 |
| CHB4 | AT1G21700 |
| CHR9 | AT1G03750 |
| CHR35 | AT2G16390 |
| CONS3 | AT3G02380 |
| DNG12 | AT1G21710 |
| FLCP39 | AT3G02310 |
| FLCP16 | AT2G22630 |
| FLCP9 | AT2G22540 |
| GTI1 | AT2G22720 |
| HMGB4 | AT2G17560 |
| JMJ27 | AT4G00990 |
| NFA1 | AT4G26110 |
| SDG23 | AT2G22740 |
| SDG37 | AT2G17900 |
| YDG2 | AT2G18000 |
| HON3 | AT2G18050 |

## Quantification per ecotype and environment

After filtering (see 'Filter for gene expression quantification'), there were 499 RNA-Seq libraries left for analysis. Next, we merged libraries per ecotype and environment, yielding 323 unique merged RNA-Seq libraries for a unique ecotype and environment (160 in 10℃, 163 in 16℃).

## Estimation of library size and abundance estimates

We followed the low level normalization proposed by *Anders and Huber (2010)*, jointly applied to the set of expression estimates across ecotypes and environmental backgrounds. First, we estimated effective library sizes as the median expression estimates across all genes. Based on this, we derived correction factors to adjust individual libraries for differences in size.

## RKPM values

Library-size adjusted raw counts were used to obtain standard read counts per million expression estimates for each gene.

# MethylC-seq data processing

## Read mapping

Reads were aligned as previously described (*Dinh et al., 2012*) to a modified pseudo-reference chromosome in which SNPs were inserted into the TAIR10 reference genome using NextGenMap (version 0.4.3; *Sedlazeck et al., 2013*) allowing up to 10% mismatch between the reads (-i 0.90) and the reference sequence and discarding reads that map equally well to more than one genomic location or have less than 45 nucleotides mapping without error to the reference sequence (-R 45). Average coverage was 12.6 X.

To correct for sample or data mix-ups, the raw data was also aligned to the first chromosome of the Columbia-0 TAIR reference genome as described above and SNP calling performed using the BISsnp package (*Liu et al., 2012*). The polymorphism data were then compared to data from genome sequencing (1001genomes.org). Accessions that did not have the highest similarity to the expected genotype were excluded from further analysis.

## DNA methylation analysis

Methylation was estimated individually for each cytosine using a python script provided with the BSMAP software package (*Xi and Li, 2009*). Conversion efficiency was estimated from the fraction of methylated cytosines in chloroplasts using the R software package (www.r-project.org, version 2.15.2). After eliminating one outlier, the samples had conversion efficiencies ranging from 99.25%–99.80% (mean = 99.59%). Genome wide average methylation levels were calculated separately for the CG, CHG and CHH contexts. The average variance between 11 biological replicates was 2.2%, 3.2% and 7.3% for CG, CHG and CHH methylation respectively, while for identical genotypes grown at different temperatures (111 pairs) CG, CHG and CHH methylation variance was 2.7%, 4.6% and 15.9% respectively. The variance in genome wide methylation levels for the 152 accessions grown at 10℃ was respectively 7.6%,

**Table 5**. Genes within 100 kb of the 15 SNPs associated with GBM at 5 or more genes whose expression is also correlated with the SNP

| SNP | Locus | Desciption | p-value |
|---|---|---|---|
| 1_19755967 | AT1G53030 | Encodes a copper chaperone | 4.72E-07 |
| 1_19755967 | AT1G52880 | **NO APICAL MERISTEM (NAM) Transcription factor with a NAC domain** | 5.47E-07 |
| 1_19755967 | AT1G52990 | Thioredoxin family protein | 2.36E-05 |
| 1_19755967 | AT1G52780 | Protein of unknown function (DUF2921) | 1.46E-04 |
| 1_4405103 | AT1G12750 | RHOMBOID-like protein 6 (RBL6); FUNCTIONS IN: serine-type endopeptidase activity | 3.74E-08 |
| 1_4405103 | AT1G12790 | RuvA domain 2-like | 2.76E-05 |
| 1_4405103 | AT1G12730 | GPI transamidase subunit | 2.81E-05 |
| 1_4405103 | AT1G13080 | CYTOCHROME P450 FAMILY 71 SUBFAMILY B POLYPEPTIDE 2 (CYP71B2) | 1.65E-04 |
| 1_7614101 | AT1G21790 | TRAM LAG1 and CLN8 (TLC) lipid-sensing domain containing protein | 1.10E-05 |
| 1_7614101 | AT1G21900 | Encodes an ER-localized p24 protein | 8.81E-05 |
| 1_7614101 | AT1G21760 | **F-BOX PROTEIN 7 (FBP7) putative translation regulator in temperature stress response** | 8.54E-04 |
| 1_912291 | AT1G03660 | Ankyrin-repeat containing protein | 1.26E-10 |
| 1_912291 | AT1G03770 | **RING1B protein with similarity to polycomb repressive core complex1 (PRC1)** | 5.76E-07 |
| 1_912291 | AT1G03940 | HXXXD-type acyl-transferase family protein | 1.18E-06 |
| 1_912291 | AT1G03610 | Protein of unknown function (DUF789) | 6.91E-06 |
| 1_912291 | AT1G03580 | Pseudogene with weak similarity to ubiquitin-specific protease 12 | 1.29E-05 |
| 1_912291 | AT1G03830 | Guanylate-binding family protein | 3.50E-05 |
| 2_6998631 | AT2G16340 | Unknown protein | 1.35E-08 |
| 2_6998631 | AT2G16210 | **Transcriptional factor B3 family protein** | 1.69E-04 |
| 2_7666059 | AT2G17630 | Pyridoxal phosphate (PLP)-dependent transferases superfamily protein | 2.47E-18 |
| 2_7660469 | AT2G17620 | Cyclin B2;1 (CYCB2;1) | 9.68E-07 |
| 2_7655016 | AT2G17740 | Cysteine/Histidine-rich C1 domain family protein | 1.22E-04 |
| 2_7655016 | AT2G17420 | NADPH-DEPENDENT THIOREDOXIN REDUCTASE 2 (NTR2) | 9.96E-04 |
| 2_7666059 | AT2G17430 | MILDEW RESISTANCE LOCUS O 7 (MLO7) | 7.56E-04 |
| 2_7915712 | AT2G18100 | Protein of unknown function (DUF726) | 1.73E-06 |
| 2_7915712 | AT2G17980 | ATSLY member of SLY1 Gene Family | 1.33E-05 |
| 2_7915712 | AT2G18400 | Ribosomal protein L6 family protein | 1.26E-04 |
| 2_7915712 | AT2G18150 | Haem peroxidase | 8.05E-04 |
| 2_7915712 | AT2G18050 | **HISTONE H1-3 (HIS1-3)** | 9.47E-04 |
| 2_9382495 | AT2G22260 | HOMOLOG OF *E. COLI* ALKB (ALKBH2) enzyme involved in DNA methylation damage repair | 1.21E-08 |

*Table 5. Continued on next page*

*Table 5. Continued*

| SNP | Locus | Desciption | p-value |
|---|---|---|---|
| 2_9382495 | AT2G21850 | Cysteine/Histidine-rich C1 domain family protein | 5.38E-06 |
| 2_9382495 | AT2G22240 | MYO-INOSITOL-1-PHOSPHATE SYNTHASE 1 (MIPS1) | 8.71E-05 |
| 2_9382495 | AT2G21940 | SHIKIMATE KINASE 1 (ATSK1) localized to the chloroplast | 1.80E-04 |
| 2_9653878 | AT2G22660 | Protein of unknown function (duplicated DUF1399) | 2.22E-14 |
| 2_9653878 | AT2G22900 | Galactosyl transferase GMA12/MNN10 family protein | 5.08E-09 |
| 2_9653878 | AT2G22830 | Squalene epoxidase 2 (SQE2) | 3.91E-06 |
| 2_9653878 | AT2G22640 | BRICK1 (BRK1) | 6.17E-05 |
| 2_9653878 | AT2G22540 | **SHORT VEGETATIVE PHASE (SVP) Floral repressor involved in thermosensory pathway** | 2.46E-04 |
| 2_9653878 | AT2G22570 | NICOTINAMIDASE 1 (NIC1) | 2.67E-04 |
| 2_9653878 | AT2G22770 | **NAI1 Transcription factor** | 7.71E-04 |
| 3_419309 | AT3G02220 | Protein of unknown function (DUF2039) | 2.06E-16 |
| 3_419309 | AT3G02230 | REVERSIBLY GLYCOSYLATED POLYPEPTIDE 1 (RGP1) | 4.58E-14 |
| 3_419309 | AT3G02300 | Regulator of chromosome condensation (RCC1) family protein | 1.25E-10 |
| 3_419309 | AT3G02120 | Hydroxyproline-rich glycoprotein family protein | 1.81E-09 |
| 3_419309 | AT3G01980 | Short-chain dehydrogenase/reductase (SDR) | 3.91E-09 |
| 3_419309 | AT3G02370 | Unknown protein | 4.53E-08 |
| 3_419309 | AT3G02020 | ASPARTATE KINASE 3 (AK3) | 4.18E-07 |
| 3_419309 | AT3G02160 | **Bromodomain transcription factor** | 2.60E-06 |
| 3_419309 | AT3G02390 | Unknown chloroplast protein | 5.60E-06 |
| 3_419309 | AT3G02050 | K+ UPTAKE TRANSPORTER 3 (KUP3) | 1.28E-05 |
| 3_419309 | AT3G02125 | Unknown chloroplast protein | 2.12E-05 |
| 3_419309 | AT3G02200 | Proteasome component (PCI) domain protein | 1.16E-04 |
| 3_419309 | AT3G02180 | SPIRAL1-LIKE3 Regulates cortical microtubule organization | 4.56E-04 |
| 3_419309 | AT3G02250 | O-fucosyltransferase family protein | 5.31E-04 |
| 3_419309 | AT3G02110 | Serine carboxypeptidase-like 25 (scpl25) | 6.18E-04 |
| 4_13290034 | AT4G26255 | **Non-coding RNA of unknown function** | 1.67E-13 |
| 4_13290034 | AT4G26450 | WPP DOMAIN INTERACTING PROTEIN 1 (WIP1) | 1.13E-04 |
| 4_13290034 | AT4G26230 | Ribosomal protein L31e family protein | 1.74E-04 |
| 4_13290034 | AT4G26160 | ATYPICAL CYS HIS RICH THIOREDOXIN 1 (ACHT1) | 5.72E-04 |
| 4_519982 | AT4G01090 | Protein of unknown function (DUF3133) | 1.23E-06 |
| 4_519982 | AT4G01230 | Reticulon family protein | 2.33E-05 |
| 4_519982 | AT4G01410 | Late embryogenesis abundant (LEA) hydroxyproline-rich glycoprotein family | 5.44E-05 |
| 4_519982 | AT4G01330 | Serine/threonine-protein kinase | 2.22E-04 |
| 4_519982 | AT4G01200 | Calcium-dependent lipid-binding (CaLB domain) family protein | 3.93E-04 |

*Table 5. Continued*

| SNP | Locus | Desciption | p-value |
|---|---|---|---|
| 4_519982 | AT4G01390 | TRAF-like family protein | 3.99E-04 |
| 4_519982 | AT4G01040 | Glycosyl hydrolase superfamily protein | 5.66E-04 |
| 4_519982 | AT4G01000 | Ubiquitin-like superfamily protein | 8.55E-04 |

9.2% and 13.2% for CG, CHG and CHH methylation, while for the 121 accessions grown at 16°C genome wide CG, CHG and CHH methylation varied 8.5%, 9.5% and 14.3% respectively.

The Bioconductor package Repitools (version 0.6.0; *Statham et al., 2010*) was used to average methylation over genomic features of interest (e.g., all genes, all transposons over 4 kB or a subset of transposons of interest). Pairwise DMRs were called individually for each accession using the R software package methylKit (version 0.5.6; *Akalin et al., 2012*) using a window size of 100 bp, an FDR rate of 0.05 and a minimum fold change of 0.3. Overlap of DMRs with (TAIR10) genomic features such as transposons and genes was calculated using the Bioconductor package ChIPpeakAnno (version 2.8.0; *Zhu et al., 2010*). For each accession, methylation data was smoothed independently for each context using the Bioconductor package BSmooth (version 0.4.5; *Hansen et al., 2012*) using the default settings. Average methylation was then calculated for (overlapping) 200 bp sliding windows centered every 100 bp across the genome. Further analysis was limited to the 200,000 windows showing the most variance among accessions.

## Population genetic analysis
### GWAS
Linear mixed models that correct for confounding by the genetic background using a kinship matrix calculated from genetic data were used throughout (*Kang et al., 2010*; *Segura et al., 2012*). To examine the effect of genotype on local CHH methylation variation, DMRs were defined by filtering the 200 bp methylation windows to remove those containing missing data (no coverage) in one or more accessions, then selecting the $10^5$ remaining windows with the greatest variance in DNA methylation. For GBM, genes were filtered to remove those that had more than 0.05 average CHG methylation or less than 0.05 average CG methylation across the accessions (*Figure 6—figure supplement 2*).

### Variance component analysis
To investigate the relative contributions of genetic and environmental effects to methylation differences we used LIMIX (*Lippert et al., 2014*), which efficiently estimates variance components using linear mixed models.

For each DMR, we considered a linear mixed model with a fixed effect for the environment and random effects for the contributions from *cis* and *trans* genetic variants and variants from the CMT2 locus. Indicating with N and E respectively the number of samples and environments (E = 2), the NxE multivariate phenotype **Y** can be written as

$$\mathbf{Y} = \mathbf{1}_{N,1}\mu^T + \mathbf{U}^{CMT2} + \mathbf{U}^{cis} + \mathbf{U}^{trans} + \psi,$$

where $\mu$ is a Ex1 vector of environment-specific mean values, and

$$\mathbf{U}^{CMT2} \sim MVN\left(\mathbf{0}, \mathbf{C}^{CMT2}, \mathbf{R}^{CMT2}\right), \mathbf{U}^{cis} \sim MVN\left(\mathbf{0}, \mathbf{C}^{cis}, \mathbf{R}^{cis}\right),$$

$$\mathbf{U}^{trans} \sim MVN\left(\mathbf{0}, \mathbf{C}^{trans}, \mathbf{R}^{trans}\right), \psi \sim MVN\left(\mathbf{0}, \Sigma, \mathbf{I}_N\right),$$

where MVN(**0**,**C**,**R**) denotes a matrix normal distribution with mean **0**, column covariance matrix **C** and row covariance matrix **R**. **R**^cis and **R**^trans indicate the genetic relatedness matrices based on cis and trans variants respectively, where all variants within 50 kb from the DMR were defined as *cis*-acting and all others as *trans*–acting. Similarly, **R**^CMT2 denotes the genetic relatedness matrix based on genotypes at the CMT2 locus. The row covariance of the noise component $\mathbf{I}_N$ corresponds to an N x N identity matrix.

The covariance matrices **C**^CMT2, **C**^cis, **C**^trans and Σ describe phenotypic correlations across environments due to these contributions, and were estimated from the data using maximum

likelihood. For each DMR, we considered up to 10 random restarts for the optimization and stopped as soon as convergence was achieved. DMRs for which no convergence was achieved were discarded from genome-wide summary statistics.

Once the model parameters have been estimated, the variance explained by environment can be calculated from $\mu$, while environment-persistent and environment-specific effects from a given random effect can be estimated by decomposing the corresponding trait covariance into a shared and an independent component (*Lippert et al., 2014*).

### QTL mapping

MethylC-seq data for the 113 F2 individuals was mapped as described in section 'Read mapping' to the Columbia-0 TAIR reference genome. SNP-calling was done directly from the methylC-seq data using the BIS-SNP package (*Liu et al., 2012*). From these SNPs local haplotype was inferred for sequential 500 Mb windows which were then used to create a haplotype map using the R package R/qtl (*Broman et al., 2003*). Mapping was done using Haley-Knot regression (*Arends et al., 2010*) with a 4 centimorgan steps size. Genome wide significance was estimated by permutation testing (1000 permutations).

### DMR calling on DNA methylation mutants

Pairwise DMRs were called for T-DNA mutants vs the wild-type control using the R software package methylKit (version 0.5.6; *Akalin et al., 2012*) using a window size of 100 bp, an FDR rate of 0.05 and a minimum fold change of 0.3. Overlap between these DMRs and 'temperature DMRs" calculated for the accessions was calculated and significance testing (Fisher's exact test) was calculated using R software.

### *Qst-Fst* test

*Fst* was computed using the Hudson estimate as suggested in *Bhatia et al. (2013)*. We note that our estimate of 0.187 is consistent with the recent estimate of *Huber et al. (2014)* (although the samples only overlap in part). For *Qst*, we first estimated northern, southern, and overall additive variance using the Hasemann-Elston regression, and a SNP-based identity-by-state matrix (*Chen, 2014*), then calculated *Qst* as $\sigma_B^2/(\sigma_B^2 + 2\sigma_w^2)$, where $\sigma_w^2$ is the weighted average of variance within north and south populations, and $\sigma_B^2$ is the variance between populations, obtained by subtracting $\sigma_w^2$ from the overall additive variance.

## Acknowledgements

This work was supported multiple sources: the National Human Genome Research Institute of the US National Institutes of Health (P50HG002790 to MN and RMC; PIS Tavaré); the European Research Council (268962 MAXMAP to MN, Marie Curie FP7 fellowship 253524 to OS); the European Community Framework Programme 7 (283496 transPLANT to MN); the National Institutes of Health Genetics (Training Grant GM07464) to EJO; the Austrian Science Fund (FWF M1369) to MJD; as well as core funding at the GMI. Sequencing was carried out by the Epigenome Center at: the University of Southern California; BGI; and the Campus Support Facilities at the Vienna Biocenter. The authors wish to thank: Andrew Smith for preliminary analyses and effectively supervising the project during the transition of MN from USC to GMI; J Bergelson for providing seed; many members of MN's lab for discussion (in particular D Filiault, F Rabanal, P Novikova, E Kerdaffrec and Ü Seren for help with various analysis tasks); A Hancock and C Huber for discussions about selection; F Sedlazeck, and P Rescheneder for advice on alignment; and F Berger and O Mittelsten Scheid for discussions and comments on the paper.

## Additional information

### Funding

| Funder | Grant reference | Author |
| --- | --- | --- |
| National Human Genome Research Institute (NHGRI) | P50HG002790 | Richard M Clark, Magnus Nordborg |
| European Research Council (ERC) | 268962 | Magnus Nordborg |

| Funder | Grant reference | Author |
|---|---|---|
| European Commission | Marie Curie FP7 fellowship 253524 | Oliver Stegle |
| European Commission | European Community Framework Programme 7 283496 | Magnus Nordborg |
| National Institutes of Health (NIH) | GM07464 | Edward J Osborne |
| Austrian Science Fund (FWF) | M1369 | Manu J Dubin |

The funders had no role in study design, data collection and interpretation, or the decision to submit the work for publication.

### Author contributions
MJD, Analysis and interpretation of data, Drafting or revising the article; PZ, Acquisition of data, Analysis and interpretation of data; DM, GJ, BV, QS, Analysis and interpretation of data; M-SR, JJ, SI, VV, Acquisition of data; EJO, FPC, PD, AK, GR, OS, Analysis and interpretation of data, Contributed unpublished essential data or reagents; QL, Provided pre-publication access to genome sequencing data; RMC, Conception and design, Contributed unpublished essential data or reagents; MN, Conception and design, Analysis and interpretation of data, Drafting or revising the article

### Author ORCIDs
Magnus Nordborg, [iD] http://orcid.org/0000-0001-7178-9748

# Additional files

### Supplementary files
• Supplementary file 1. Multiple sequence alignment of CMT2 sequences from different accessions.

• Supplementary file 2. Multiple sequence alignment of CMT2 sequences from different accessions in FASTA format.

• Supplementary file 3. Source for scripts used to extract SNPs from RNA-seq data and comparing to existing 250k SNP data.

### Major dataset
The following datasets were generated:

| Author(s) | Year | Dataset title | Dataset ID and/or URL | Database, license, and accessibility information |
|---|---|---|---|---|
| Dubin MJ, Nordborg M | 2014 | DNA methylation variation in Arabidopsis has a genetic basis and appears to be involved in local adaptation | http://www.ncbi.nlm.nih.gov/geo/query/acc.cgi?acc=GSE54292 | Publicly available at NCBI Gene Expression Omnibus GSE54292. |
| Osborne EJ, Zhang P, Remigereau MS, Drewe P, Rätsch G, Nordborg M, Clark RM | 2014 | DNA methylation variation in Arabidopsis has a genetic basis and appears to be involved in local adaptation | http://www.ncbi.nlm.nih.gov/geo/query/acc.cgi?acc=GSE54680 | Publicly available at NCBI Gene Expression Omnibus GSE54680. |
| Dubin MJ, Nordborg M | 2014 | Population epigenomics of Swedish Arabidopsis | http://www.ncbi.nlm.nih.gov/geo/query/acc.cgi?acc=GSE65685 | Publicly available at NCBI Gene Expression Omnibus GSE65685. |
| Dubin MJ, Nordborg M | 2014 | Effect of Chromomethylase-2 and Dicer-like-3 knockouts on DNA methylation | http://www.ncbi.nlm.nih.gov/geo/query/acc.cgi?acc=GSE66017 | Publicly available at NCBI Gene Expression Omnibus GSE66017. |

| Author(s) | Year | Dataset title | Dataset ID and/or URL | Database, license, and accessibility information |
|---|---|---|---|---|
| Long Q, Rabanal FA, Meng D, Huber CD, Farlow A, Platzer A, Zhang Q, Vilhjálmsson BJ, Korte A, Nizhynska V, Voronin V, Korte P, Sedman L, Mandáková T, Lysak MA, Seren Ü, Hellmann I, Nordborg M | 2013 | Massive genomic variation and strong selection in Arabidopsis thaliana lines from Sweden | http://plone.gmi.oeaw.ac.at/downloads/nordborg/data-release-for-massive-genomic-variation-and-strong-selection-in-arabidopsis-thaliana-lines-from-sweden | Publicly available at Gregor Mendel Institute of Molecular Plant Biology. |

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
