## [Decision Letter]

Thank you for sending your work entitled “DNA methylation variation in Arabidopsis has a genetic basis and shows evidence of local adaptation” for consideration at *eLife*. Your article has been favorably evaluated by Stylianos Antonarakis (Senior editor), a Reviewing editor, and three reviewers..

The Reviewing editor and the other reviewers discussed their comments before we reached this decision, and the Reviewing editor has assembled the following comments to help you prepare a revised submission.

The work was considered important and novel and no major technical concerns were raised.

Below are the key issues to be addressed in revision:

1) What is the significance of the identified variation in CMT2-dependent CHH methylation (in contrast to others species like maize that don't have this enzyme) and the same for the GBM variation? Do these results reflect differences in chromatin compaction? What does an increase in ∼10% gene body methylation actually mean? Is there cell type specificity? Is the CG methylation higher at each CG site or are more CG sites methylated or is it a combination of both? What is the explanation for differential CHH between 10 and 16C?

2) The present data does not show that the levels of DNA methylation influence local adaptation. Rather, it shows that variation in methylation has a strong genetic basis. Furthermore, there is no evidence showing that this genetic variation underlies local adaptation. The statement about adaptation should be toned down or removed.

---

## [Author Response]

*Below are the key issues to be addressed in revision*:

1) What is the significance of the identified variation in CMT2-dependent CHH methylation (in contrast to others species like maize that don't have this enzyme)…

CMT2 appears to be present in most plants, with the notable exception of maize (at least for the reference strain), suggesting it plays an important role. It has been suggested (Shen, 2014) that differences in CMT2 dependent CHH methylation affect tolerance to thermal stress (which would certainly fit with maize being very tolerant to heat stress). It could also potentially affect TE expression/expansion, although we have been unable to find evidence for this in our data. We now mention these possibilities in the text.

… and the same for the GBM variation?

The function of GBM is generally poorly understood, but some involvement in transcriptional regulation seems highly likely (GutierrezArcelus, 2013). In other data, the variation in GBM between north and south is associated with differences in the level as well as variance of expression and is also strongly correlated with climate. It maps to loci that have previously been shown to be involved in local adaptation to climate. To strengthen this further, we have added an FstQst analysis that demonstrates that the trait is under divergent selection. A deeper understanding of the mechanisms involved will have to await the cloning and characterization of the transacting loci affecting GBM.

Do these results reflect differences in chromatin compaction?

Given that most of the variation in DNA methylation was fairly mild (max ∼10%) we consider it unlikely that they cause largescale changes in chromatin organization, especially when one considers that very few null mutants involved in DNA methylation (with a few notable exceptions such as mom1, ddm1 and met1) seem to affect chromatin compaction. That said, natural variation in chromatin compaction clearly does exist, at least in the Cvi accession (Tessodori, 2009) and we have mentioned the possibility. Future studies should investigate this.

What does an increase in ∼10% gene body methylation actually mean? Is there cell type specificity? Is the CG methylation higher at each CG site or are more CG sites methylated or is it a combination of both?

While DNA methylation variation between cell types has been described in the endosperm and pollen (for example Calarco, 2012) and also the germline (Baubec, 2014), in all of these cases it seems to be limited to transposons and mostly affect CG methylation. Furthermore, several investigations have shown that CG gene body methylation is very highly conserved between different organs (root, shoot, rosette, cauline leaf etc; ColemanDerr & Zilberman, 2012; Seymore DK, 2014). Given that our samples were all rosettes sampled at the same developmental stage, cell type should have very little if any affect on GBM. We have expanded our analysis to demonstrate that roughly the same number of CG sites are methylated in north and south, with the difference being a skew towards higher values in the north (Figure 6—figure supplement 3 and Figure 6—figure supplement 4).

What is the explanation for differential CHH between 10 and 16C?

The difference appears to affect primarily RdDM target sites. Given that the temperature sensitivity of the smallRNA processing pathway is well documented (see, e.g., Romon, 2013), this would seem to be the most plausible explanation. This is noted in the text.

*2) The present data does not show that the levels of DNA methylation influence local adaptation. Rather, it shows that variation in methylation has a strong genetic basis. Furthermore, there is no evidence showing that this genetic variation underlies local adaptation. The statement about adaptation should be toned down or removed*.

Here we must respectfully disagree. It is certainly correct that we have no evidence that DNA methylation is somehow important for adaptation independent of genetic variation, and indeed we show that it has a strong genetic basis. However, it is wrong to say that there is no evidence for selection. The variants affecting GBM are strongly overrepresented in sweep regions, and show precisely the kinds of climate associations that would be expected if they were involved in local adaption. Furthermore, the northsouth divergence in GBM is far too great to be due to genetic drift (we have added an FstQst analysis to make this clearer). Of course the evidence is all indirect, and of course all these analyses come with significant caveats, but this is usually the case in population genetics. Generating direct evidence of fitness differences is extremely difficult and time consuming.

We are well aware that a number of strange claims concerning the role of DNA methylation are currently being made, and it precisely for this reason that we try to make it very clear that we are talking about genetic effects in this paper (DNA methylation is essentially treated as a phenotype).